# Distributed Fusion of Sensor Data in a Constrained Wireless Network

**DOI:** 10.3390/s19051006

**Published:** 2019-02-27

**Authors:** Charikleia Papatsimpa, Jean-Paul Linnartz

**Affiliations:** 1SPS group, Eindhoven University of Technology, 5612 AZ Eindhoven, The Netherlands; j.p.linnartz@tue.nl; 2Signify Research, 5656 AE Eindhoven, The Netherlands; j.p.linnartz@signify.com

**Keywords:** Internet of Things (IoT), sensor fusion, smart building, efficient transmission, wireless sensor networks

## Abstract

Smart buildings with connected lighting and sensors are likely to become one of the first large-scale applications of the Internet of Things (IoT). However, as the number of interconnected IoT devices is expected to rise exponentially, the amount of collected data will be enormous but highly redundant. Devices will be required to pre-process data locally or at least in their vicinity. Thus, local data fusion, subject to constraint communications will become necessary. In that sense, distributed architectures will become increasingly unavoidable. Anticipating this trend, this paper addresses the problem of presence detection in a building as a distributed sensing of a hidden Markov model (DS-HMM) with limitations on the communication. The key idea in our work is the use of a posteriori probabilities or likelihood ratios (LR) as an appropriate “interface” between heterogeneous sensors with different error profiles. We propose an efficient transmission policy, jointly with a fusion algorithm, to merge data from various HMMs running separately on all sensor nodes but with all the models observing the same Markovian process. To test the feasibility of our DS-HMM concept, a simple proof-of-concept prototype was used in a typical office environment. The experimental results show full functionality and validate the benefits. Our proposed scheme achieved high accuracy while reducing the communication requirements. The concept of DS-HMM and a posteriori probabilities as an interface is suitable for many other applications for distributed information fusion in wireless sensor networks.

## 1. Introduction

Smart buildings are becoming a reality thanks to the availability of low-cost, easy to install Internet of Things (IoT) devices, such as sensors and actuators. An extensive amount of research has been dedicated to developing occupancy-based control systems that exploit information on user presence to dynamically adjust energy-related appliances and building systems (HVAC, lighting, or other appliances). Those systems are based on a network of IoT-enabled sensor devices that continuously monitor the space with the aim to provide real-time information on user occupation. For example, SCOPES, a distributed smart cameras object position estimation system [1], uses real-time occupancy data to create predictive occupancy models [2]. Those models can be integrated into a building conditioning system for usage-based demand control conditioning strategies, most notably HVAC and lighting. Other approaches, as in [3,4], achieve relatively accurate localization from sensor badges or smart phones to enable building energy control. Advanced sensor modalities, such as cameras, or wearable devices, such as smartphones, increase the accuracy of the system yet, at the same time, increase cost and intrusiveness. Simple, wireless, binary sensors are preferred, which are easy to retrofit in existing buildings and comply with the existing privacy regulations. 

To accommodate growing expectations, sensing systems need to interact and combine information to reduce sensor uncertainty while reducing cost and keeping the sensor infrastructure non-intrusive. This forms an intrinsically multi-sensory data fusion problem, in which the readings from multiple sensors must be combined into a coherent structure. For example, information collected by a network consisting of traditional PIR sensors is fused in a probabilistic framework based on the Bayesian probability theory in [5]. Nguyen and Aiello [6] combined observations from a wireless network of simple sensors (infrared, pressure, and acoustic) in a recognition system that performs indoor activity recognition. Sensor data were periodically pushed to a central base station, where, an algorithm determined activity by associating the data with a specific configuration. Similarly, Huang and Mao [7] proposed a hybrid detection method using the combination of CO_2_ and light sensors by which the measurement results of CO_2_ and light levels are transmitted to a central control computer via wireless communication. 

Yet, most of the smart building applications found in the literature, including [8], make use of centralized architectures with sensing nodes exchanging or delivering their readings into a data sink (e.g., a base station). However, this sensor interaction is constrained by the limited resources of sensor nodes, such as limited memory, battery power, and computation and communication capabilities. Especially given that the number of interconnected devices is exhibiting an exponential rise towards a forecasted 50 billion connected devices by 2020 [9], distributed architectures are becoming more relevant than ever. The interconnected devices will generate a vast amount of data; thus, devices may be required to make some decisions locally. Edge analytics will become necessary to determine which data is worth sending and when or whether efficient data compression or aggregation into meaningful information can be performed.

Motivated by this, we propose a distributed architecture for smart buildings. We approach the problem of presence detection as a Markov process that moves between a number of states, e.g., presence or absence. Our challenge is to determine the current state of that process from the sequence of noisy observations made by (imperfect) sensors. This concept is known as a hidden Markov model (HMM) and has been extensively explored for other applications, but we extend this to allow distributed sensing and distributed processing. We thus introduce a distributed approach in which, spatially separated sensor nodes try to effectively combine their observations to estimate the state of the underlying process. In fact, in a centralized architecture, a theoretically optimal solution for a HMM with multiple sensors, can be implemented [8], but it requires perfect and frequent communication between nodes. Yet, specifically for IoT applications based on wireless sensor networks, this is prohibitive. The large amount of bandwidth required to send raw sensor observations through the network at low latency is a significant disadvantage that drains the limited battery resources (the power consumed by inter-sensor communication is the main source of battery consumption [10,11]) and increases traffic load. We now investigate whether we can reduce communication, while maintaining performance. For this distributed-HMM (DS-HMM) architecture, observations are initially processed locally and independently of other nodes, and we switch from exchanging the latest observations towards conditionally exchanging local likelihood ratio values. We propose a set of fusion algorithms to merge data from various HMMs running separately on all sensor nodes. We model a non-homogeneous HMM with time-of-day dependent transition rates, because activity patterns change during the day. We build a simple experimental prototype system that acts as an initial proof-of-concept of the feasibility of the suggested solution and test functionality. The aim is to verify to what extent our solution is robust in the real conditions of a wireless sensor network and with human behavior that may not be captured exactly in a Markov process.

## 2. System Overview

### 2.1. Overview

In this section, we approach the problem of monitoring room occupancy as a distributed hidden Markov model, in which *K* spatially separated sensor nodes try to effectively combine their observations to estimate the state of the underlying process. In order to lower the communication load and thereby maximize the lifetime of the wireless sensor network, sensor interaction needs to be limited. Minimizing data transmissions is also important to avoid collisions among concurrent transmissions. This section describes in detail the occupancy model of the local nodes, the proposed transmission policy, and the fusion algorithms developed to effectively combine local information.

### 2.2. Prerequisite: Occupancy Inference based on HMMs

Hidden Markov models (HMMs) and their extensions are a particular representation of dynamic graphical models (DGMs), popular for modeling time series data. They offer a natural approach to encode causality (conditional independency) and provide a principled framework for combining prior knowledge and data. Therefore, the HMM framework can provide a computationally efficient and sufficiently expressive solution to the problem of office occupancy. We model human presence as a Markovian on–off process with two possible user states: $q_{t}=0$ represents the state in which the user is absent, while $q_{t}=1$ represents the state in which user is present. However, our HMM approach is not principally limited to two states [12], and the architecture described here can also be extended to a larger state space. The possible user states are unknown (hidden) to the system but can only be observed through an imperfect sensor network, which we interpret as another set of stochastic processes that produce the sequence of observations. The resulting system forms a hidden Markov model that requires the specification of the following set of parameters [13]:(1)State transition probability matrix: $A=\left\{a_{ij}\right\},{\;a}_{ij}=P\left(q_{t}=j|q_{t-1}=i\right)$. The transition probabilities describe how space occupancy changes over time. Because we assume only two possible states, two transition probabilities need to be specified, namely $P\left(q_{t}=1|q_{t-1}=0\right)$ and $P\left(q_{t}=0|q_{t-1}=1\right)$.(2)Emission probability matrix: $B=\left\{b_{ij}\right\},b_{i}\left(j\right)=P\left(r_{t}=j|q_{t}=i\right)$. The observed symbols are sensor readings monitoring the hidden states, such as, for example, ultrasound or PIR sensors providing measurements. Our mathematical approach is very suitable to combine data from sensors with different reliabilities. Yet, in the examples that we give in this work, we use one type of sensor, namely ultrasound (USR) sensors. Typically, sensor measurements are continuous-valued variables, such as a time-of-flight distance. However, for simplification in the calculations, we map each continuous observation to a binary value $r_{t}\in \left\{0,\;1\right\}$. The emission probabilities are a metric of the quality of the sensor modality used and thus are directly linked to the successful detection rate (SDR) and the false alarm rate (FAR). The SDR is the probability of obtaining a sensor reading given that a person is present, while the FAR is the probability of obtaining a sensor reading given that a person is absent. Accordingly, the emission probability matrix is defined as follows:(1)$$B=\left(\begin{array}{cc} \;b_{0}\left(0\right)=1-\;b_{0}\left(1\right) & \;b_{0}\left(1\right)=FAR\\ \;b_{1}\left(0\right)=1-\;b_{1}\left(1\right) & \;b_{1}\left(1\right)=SDR \end{array}\right).$$(3)$\pi =\left\{\pi_{i}\right\}$: Initial state probability vector. The initial state distribution specifies the occupancy probability at the initial time step t=0, prior to any observation. Yet, in our application, the influence of the initial state rapidly vanishes.

The hidden Markov model parameters λ={A,B,π} can be estimated theoretically or experimentally, using labeled sequences of observations and states in a training step.

### 2.3. Dynamic Transition Probabilities

In the typical HMM model formulation, the transition probabilities $A=\left\{a_{ij}\right\}$ are considered to be constant over time, i.e., A(t)=A. Yet, as we addressed in [14], the time of the day has a significant influence on the probability of occupancy. For instance, because it is much more probable that someone is present only during the day, a sensor trigger at night is more likely to be a false alarm than a daytime trigger. This leads to the non-stationary behavior of the transition probabilities. The non-homogeneous model exploits this prior knowledge by mathematically biasing the HMM probabilities. Thus, we extended the HMM model framework to allow for non-homogeneous transition probabilities that depend on the time of day. Specifically, we let the transition probabilities $a_{01,\;t}$ and $a_{10,\;t}$ vary over time, while the two remaining probabilities are set to $a_{00}\left(t\right)=1-a_{01,\;t}$ and $a_{11}\left(t\right)=1-a_{10,\;t}$. We specified the transition probabilities as
(2)$$a_{01,\;t}=\Lambda_{1,t}\;a_{10,\;t}=\Lambda_{2,t}$$
where $\Lambda_{1,t}$ and $\Lambda_{2,t}$ are design matrices corresponding to a function linking the transition probabilities to the time of day. Given that there is no theoretical basis for choosing a suitable function, one could rely on experimental data to obtain estimations of the transition probabilities of interest. We used the experimental data collected in our experimental set-up described in [14] to get an hourly estimate of the conditional probability of transiting from the absent state to the present state and vice versa. One could argue that a finer granularity makes sense, for instance to capture that meetings often start around the top of the hour. However, this time resolution would require vast amounts of data, while for our dataset and our specific university student setting, it would not give a justifiable, statistically significant result. The matrices $\Lambda_{1,t}$ and $\Lambda_{2,t}$ can be designed by applying a fitting curve over the experimental data. We chose to apply cubic Hermite polynomials. Various fitting models were tested; however, shape-preserving interpolants showed the best fit to our experimental data. Figure 1 shows the average transition probability from the absent state to the present state ($a_{01,\;t}$) as a function of the time of day over all workdays. We observed that our participants arrived at work between 8:30 and 10:00. A “valley” was, as expected, observed around 12:30 (corresponding to lunch time). A global maximum was seen around 14:00, after which it dropped and approached zero at 20:00. Although the prior probability of presence at night is very close to zero, we noticed that in our system, we needed to set an artificial minimum to prevent undesirable behavior during which the system failed to switch on lights during unlikely nightly events and incidents. This minimum value was set according to our model parameters to ensure a system reaction to a new presence within 1 s; that is, it required a different regime than the daytime compromise between comfort versus energy conservation. A similar analysis was made for the transition probability for an employee to transit from the present state to the absent state ($a_{10,\;t}$).

As a verification of whether our obtained transition probabilities were plausible, we computed the resulting time-varying state probabilities and checked whether these fit with typical office occupancy data, e.g., the ASHRAE 90.1 2004 recommended standard curves for private offices [15]. The occupancy pattern that follows is derived from the estimated transition probabilities according to
(3)$$P\left(q_{t}=1\right)=a_{11,\;\;t}P\left(q_{t-1}=1\right)+a_{01,\;\;t}P\left(q_{t-1}=0\right).$$

The comparison in Figure 2 confirms that both profiles share important characteristics, that is, occupancy starts increasing in the morning, dips at lunchtime, rises again to the same level, and drops near the end of the workday. However, in our special case, we observed a time delay in the arrival time of employees, a delay that can be justified by the more flexible working policy that is typical in a university setting. This diversity suggests that there might be statistical differences between different types of offices. Variations might also occur according to the geographical zone, type of office (private, open floor plan, etc.), suggesting that more research is necessary to develop a larger building database spanning over different geographical regions and office types. Alternatively, a self-learning algorithm can be used. 

### 2.4. State Estimation in HMMs

The goal of the presence detection system is to allow each individual sensor *k* to estimate the probability of a subject being present (or absent), considering that we have a set of sensor readings $r_{1:t}^{\left(k\right)}=r_{1}^{\left(k\right)},\;r_{2}^{\left(k\right)},..,\;r_{t}^{\left(k\right)}$ and that we can rely on the system’s memory to know the previous state estimation. In particular, we aimed to express $P\left(q_{t}=i|r_{1:t}^{\left(k\right)}\right)$, which can be written using Bayes theorem as
(4)$$P\left(q_{t}=i|r_{1:t}^{\left(k\right)}\right)=\frac{P\left(q_{t}=i,r_{1:t}^{\left(k\right)}\right)}{P\left(r_{1:t}^{\left(k\right)}\right)}$$
where i∈{0,1} was the possible user states representing absence and presence, respectively, and $r_{1:t}^{\left(k\right)}=r_{1}^{\left(k\right)},\;r_{2}^{\left(k\right)},..,\;r_{t}^{\left(k\right)}$ represents the series of observations of the *k*-th sensor up to time *t*.

Given the constructed HMM model λ={A,B,π}, the HMM framework defines the forward–backward procedure that allows us to inductively calculate the forward variable $\alpha_{t}\left(j\right)=P\left(r_{1}r_{2}\ldots r_{t},q_{t}=j|\lambda \right)$, i.e., the joint probability of the observation sequence. The forward variable gives an estimate of the most likely user state at time *t* according to
(5)$$\;\alpha_{t}\left(j\right)=b_{j}\left(r_{t}\right)\sum_{i}a_{ij}\;\alpha_{t-1}\left(i\right).$$

Because we considered only two possible user states, with the user either present or absent, in order to simplify the calculations, we introduced the likelihood ratio $Q_{t}^{\left(k\right)}$ defined as the probability ratio
(6)$$Q_{t}^{\left(k\right)}=\frac{P\left(q_{t}=1,r_{1:t}^{\left(k\right)}\right)}{P\left(q_{t}=0,r_{1:t}^{\left(k\right)}\right)}=\frac{\;\alpha_{t}\left(1\right)}{\;\alpha_{t}\left(0\right)}.$$

After some mathematical reformulation and by substituting Equation (5) in Equation (6), we derived a simple expression to inductively calculate $Q_{t}^{\left(k\right)}$ as
(7)$$Q_{t}^{\left(k\right)}=\frac{b_{1}\left(r_{t}^{\left(k\right)}\right)}{b_{0}\left(r_{t}^{\left(k\right)}\right)}\left[\frac{a_{11}Q_{t-1}^{\left(k\right)}+a_{01}}{a_{10}Q_{t-1}^{\left(k\right)}+a_{00}}\right].$$

Each individual sensor *k* runs the above HMM algorithm (Equation (7)) to calculate the local likelihood ratio $Q_{t}^{\left(k\right)}$. Based on the local HMM algorithm, each node in the network individually estimates the user state based on its local view, i.e., from the *k*-th sensor’s observations. This estimate is accompanied by a confidence level, i.e., the magnitude of the likelihood $\left|Q_{t}\right|$, or alternatively $\log\left|Q_{t}\right|$ can be interpreted as the degree of belief on the user state. In simple words, the sign of $\log\;Q_{t}$ is the hard decision on the state, and the magnitude $\log\left|Q_{t}\right|$ is the reliability of this decision.

### 2.5. Decision Rule

The decision rule determining presence/absence is given by a simple thresholding operation on the log-likelihood ratio as
(8)$$q_{t}=L\left(Q_{t}\right)=\{\begin{array}{c} 1,\;\;\mathrm{if}\;\log\;Q_{t}>\delta \\ 0,\;\;\mathrm{if}\;\log\;Q_{t}\leq \delta \end{array}$$
where δ is a preselected presence/absence threshold. Typically, $Q_{t}$ values span over several orders of magnitude. Thus, using a logarithmic form allows us to better adjust the threshold to make a tradeoff between false positives and false negatives. 

### 2.6. Communication Strategy

Occupancy detection can significantly be improved by using data fusion techniques to simultaneously utilize the information collected by multiple spatially separated IoT nodes. This synergistic use of overlapping and complementary data provides a reliability that may otherwise be unavailable from individual sources. Centralized fusion approaches, though easier to approach the optimal solution [16], in practice, give rise to problems with computational and communication bottlenecks, as they require the transmission of measurements from all nodes to a fusion center. On the other hand, in a distributed architecture, communication can be adaptive and dependent on the information content of the individual nodes. This reduces the necessary data communication, because data do not have to be sent to a central processing node, and it allows for faster access to fusion results, because there is less communication latency. Towards this direction, we propose an adaptive method to exchange data. Our proposed strategy is based on a simple intuition. We exploited the fact that $\left|\log Q_{t}\right|$ can be seen as a metric of belief on the user state and let individual nodes exchange information only when their confidence on their local estimate is low. In mathematical terms, a test node *k* communicates according to
(9)$$\left|\log Q_{t}\right|\leq \gamma \to \mathrm{send}\;\;msg_{k}$$
where $msg_{k}$ is the message that node *k* sends to all nodes in the network and γ is the chosen communication threshold. The communication threshold can be adjusted as a tradeoff between communication load and performance.

### 2.7. Data Fusion

The data fusion problem at hand consists of two main questions, namely what type of information nodes should exchange and how to optimally combine the available information in order to achieve high performance results (who, when, what, and how). In a centralized architecture with multiple nodes, each making their own noisy observations, optimal detection can be achieved if all the nodes exchange all the observations at every time unit [8]. Yet, this leads to an excessive communication load. In the case that sensor communication is constrained, it is an open question of what data nodes should exchange to reach a good solution. For the fused estimate to be the same as the optimal (centralized) estimate, the information communicated by each node has to contain all the information needed to reconstruct the optimal estimate. According to our system architecture, this includes the history of previous estimates. It requires the algorithm to trace back to the latest moment in which all the data were available and to retroactively calculate the joint likelihood ratio, posing major memory and computational requirements. The tracing-back depth (TD), i.e., how far back in time the HMM algorithm needs to reprocess measurements, might theoretically include hours of past observations, but transmitting the full history is not practical. We explored a more pragmatic approach, namely to send only a few recent observations in every message (packet) instead of the full history. In practice, the information carried by a measurement diminishes rapidly with its age, i.e., only if the measurement is not too far back in time, is it relevant enough for an update. We experimentally investigated the required length of the trace-back to maintain adequate performance as a function of the sensor reliability, expressed as the sensor SDR and FAR that define the emission probability matrix. Figure 3 shows the mean square error $e_{rms}=\sqrt{E\left({\left(\frac{\hat{Q\left(TD\right)}-Q}{Q}\right)}^{2}\right)}$ as a function of the length of the trace-back. We found a very clear dependency on the sensor reliability. In fact, for a very reliable sensor, tracing back to only TD=4, the latest observations turned out to be very close to the optimum reconstruction; however, unreliable sensors observing a quite stable state can benefit from looking back much further into history. Yet, even with a rather unreliable sensor, tracing back up to TD=10 past observations appears adequate, at least in a network with K = 2 nodes.

The algorithmic steps required for the calculation of the joint likelihood ratio with tracing-back in history can be summarized as follows (Algorithm 1):


**Algorithm 1: Distributed hidden Markov model (HMM) Algorithm with Sub-Optimal Retroactive Reconstruction**
1: Initialization: Q1(k)=b1(r1(k))π1b0(r1(k))π02: **while** new data exist **do**3: Calculate the likelihood ratio:
Qt(k)=b1(rt(k))b0(rt(k))[a11Qt−1(k)+a01a10Qt−1(k)+a00]4: **if**
|logQt|≤γ
**do**5: send msgk=(rt−TD(k),…, rt(k))6: **end if**7: **if**
msg1, …,msgK received **do**8: Trace back to Qt−TD(k)9: Retroactively calculate Qt(k∗) according to
Qt(k∗)=b1(rt(1))b0(rt(1))…b1(rt(K))b0(rt(K))[a11Qt−1(k)+a01a10Qt−1(k)+a00]
starting from Qt−TD(k).10: Update Qt(k)←Qt(k∗)11: **end if**12: Estimate state according to decision rule Qt=L(Qt)={1,  if logQt>δ0,  if logQt≤δ13: **end while**

The reconstructed state estimate $Q_{t}^{\left(k*\right)}$ is the joint likelihood ratio given information from all sensors defined as
(10)$$Q_{t}^{\left(k*\right)}=\frac{P\left(q_{t}=1,r_{1:t}^{\left(1\right)},r_{1:t}^{\left(2\right)},\ldots ,r_{1:t}^{\left(K\right)}\right)}{P\left(q_{t}=0,r_{1:t}^{\left(1\right)},r_{1:t}^{\left(2\right)},\ldots ,r_{1:t}^{\left(K\right)}\right)}.$$

The tracing-back algorithm is an attractive solution that achieves performance quite close to the centralized HMM solution with limited transmission requirements. Still, if the required tracing-back depth is large, tracing back and storing information involves lengthy calculations and storage requirements that might pose challenges for real-time processing. For a wireless sensor network, an attractive distributed data fusion method minimizes the payload of messages. In our opinion, just the exchange of likelihood ratios (LRs), and thus Q values, is more efficient and more practical as long as the number of nodes involved is not too large.

In fact, one can exploit that the nodes have already calculated a local state estimate in the form of the $Q_{t}$ ratio, that is, the ratio of the state estimation given the series of sensor observations so far (including the latest observation). This ratio captures all the necessary information that the system needs to know, suggesting that it is a suitable and sufficient variable to be exchanged. Switching from communicating raw data to communicating local likelihood estimates raises a second question, namely what is the optimum and most efficient fusion technique to combine data. To the best of our knowledge, no suitable fusion function *f* has been investigated previously for combining local estimates without requiring more than the current (latest) state estimate.

The mathematical expression representing the fusion of multiple sensor observations in an HMM framework is given as
(11)$$Q_{t}^{\left(k*\right)}=\frac{P\left(q_{t}=1|r_{1:t}^{\left(1\right)},\ldots ,r_{1:t}^{\left(K\right)}\right)}{P\left(q_{t}=0|r_{1:t}^{\left(1\right)},\ldots ,r_{1:t}^{\left(K\right)}\right)}.$$

Applying the Bayes rule gives
(12)$$Q_{t}^{\left(k*\right)}=\frac{P\left(r_{1:t}^{\left(1\right)},\ldots ,r_{1:t}^{\left(K\right)}|q_{t}=1\right)P\left(q_{t}=1\right)}{P\left(r_{1:t}^{\left(1\right)},\ldots ,r_{1:t}^{\left(K\right)}|q_{t}=0\right)P\left(q_{t}=0\right)}.$$

If we could rely on the conditional independence, we would reach
(13)$$Q_{t}^{\left(k*\right)}=\frac{P\left(r_{1:t}^{\left(1\right)}|q_{t}=1\right)}{P\left(r_{1:t}^{\left(1\right)}|q_{t}=0\right)}\ldots \;\frac{P\left(r_{1:t}^{\left(K\right)}|q_{t}=1\right)}{P\left(r_{1:t}^{\left(K\right)}|q_{t}=0\right)}\;\frac{P\left(q_{t}=1\right)}{P\left(q_{t}=0\right)}\;=\frac{P\left(q_{t}=1,r_{1:t}^{\left(1\right)}\right)}{P\left(q_{t}=0,r_{1:t}^{\left(1\right)}\right)}\ldots \;\frac{P\left(q_{t}=1,r_{1:t}^{\left(K\right)}\right)}{P\left(q_{t}=0,r_{1:t}^{\left(K\right)}\right)}\;{\frac{P\left(q_{t}=0\right)}{P\left(q_{t}=1\right)}}^{K-1}\;=Q_{t}^{\left(1\right)}\ldots Q_{t}^{\left(K\right)}c^{K-1}$$
which is a multiplication of the K individual likelihood ratios but with a K−1 times correction for the *K*-times use of the prior probability ratio *c*. This is an intuitively appealing fusion formula. However, this formula requires conditional independence of measurements, which regrettably is not a correct assumption, because in any useful sensor system, measurements at least statistically depend on the state $q_{t}$. In many situations, it is reasonable to assume, or to approximate, that observation errors are conditionally independent; thus,
(14)$$P\left(r_{t}^{\left(1\right)},\ldots ,r_{t}^{\left(K\right)}|q_{t}\right)=\prod_{k=1}^{K}P\left(r_{t}^{\left(k\right)}|q_{t}\right).$$

However, Equation (13) requires that past observations, say at t−1 are also independent. Yet, $r_{t-1}^{\left(1\right)}$ gives information on the state $q_{t-1}$ and thus influences the probability on $r_{t-1}^{\left(2\right)}$. Nonetheless, though formally sub-optimum, we explored such fusion *f* formulas, as in our case, these appeared suitable. Inspired by Equation (13), a possible fusion function *f* can be expressed as
(15)$$Q_{t}^{\left(k*\right)}=f\left(Q_{t}^{\left(1\right)}\ldots Q_{t}^{\left(K\right)}\right)=Q_{t}^{\left(1\right)}\ldots Q_{t}^{\left(K\right)}c^{K-1}.$$

Because transitions between states are balanced, that is, the number of times a person gets into a room is equal to the number of times the person leaves the room, the ratio of the prior probabilities can be expressed as
(16)$$P\left(q_{t}=0\right)a_{01}=P\left(q_{t}=1\right)a_{10}\Rightarrow \;c=\frac{a_{10}}{a_{01}}.$$

This term can be interpreted as a correction term that prevents the function from double counting the priors. The algorithmic steps required for the calculation of the joint likelihood ratio can be summarized as follows (Algorithm 2).


**Algorithm 2: Distributed HMM Algorithm with Correction Term Fusion**
1: Initialization: Q1(k)=b1(r1(k))π1b0(r1(k))π02: **while** new data exist **do**3: Calculate the likelihood ratio: Qt(k)=b1(rt(k))b0(rt(k))[a11Qt−1(k)+a01a10Qt−1(k)+a00]4: **if**
|logQt|≤γ
**do**5: send msgk=(Qt(k))6: **end if**7: **if**
msg1, …,msgK received **do**8: Calculate Qt(k∗) according to Qt(k∗)=Qt(1)…Qt(K)cK−110: Update Qt(k)←Qt(k∗)11: **end if**12: Estimate state according to decision rule qt=L(Qt)={1,  if logQt>δ0,  if logQt≤δ13: **end while**

We further explored alternative fusion formulas inspired by the field of pattern recognition, where it is well known that in many situations, combining the output of several classifiers leads to an improved classification result. Combining local estimates from *K* separate nodes can be interpreted as a problem of combining the output of *K* classifiers, each providing an estimation of the given class q (q∈{0,1}) based on the calculation of posterior probabilities given the input measurement vector. Thus, from a probabilistic point of view, we may straightforwardly conceive a weighted mixture of individual classifiers, namely
(17)$$Q_{t}^{\left(k*\right)}=f\left(Q_{t}^{\left(1\right)}\ldots Q_{t}^{\left(K\right)}\right)=\frac{1}{\sum_{k}w_{k}}\overset{K}{\underset{k=1}{\sum }}w_{k}Q_{t}^{\left(k\right)}$$
where the weights $w_{k}\;\left(k=1,\ldots ,K\right)$ are interpreted as a probability that reflects the reliability of the k-th classifier. This interpretation of weights seems to be especially appropriate when defining weights in terms of the accuracy of individual classifiers [17]. In that way, the influence of each sensor signal is weighed in the fusion scheme according to the reliability or the accuracy of its estimation. 

The algorithmic steps required for the calculation of the joint likelihood ratio can be summarized as follows (Algorithm 3).


**Algorithm 3: Distributed HMM Algorithm with Weighted Averaging Fusion**
1: Initialization: Q1(k)=b1(r1(k))π1b0(r1(k))π02: **while** new data exist **do**3: Calculate the likelihood ratio: Qt(k)=b1(rt(k))b0(rt(k))[a11Qt−1(k)+a01a10Qt−1(k)+a00]4: **if**
|logQt|≤γ
**do**5: send msgk=(Qt(k))6: **end if**7: **if**
msg1, …,msgK received **do**8: Calculate Qt(k∗) according to Qt(k∗)=1∑kwk∑i=1KwkQt(k)10: Update Qt(k)←Qt(k∗)11: **end if**12: Estimate state according to decision rule qt=L(Qt)={1,  if logQt>δ0,  if logQt≤δ13: **end while**

## 3. Proof of Concept

This section addresses the prototype that we built to act as a proof-of-concept for the proposed distributed occupancy detection approach described in Section 2. 

### 3.1. Implementation Details

As shown in Figure 4, the prototype node contains three different parts, interconnected with each other: a USR sensor model SRF08 from Davatech (https://www.robot-electronics.co.uk/htm/srf08tech.html), an Arduino Uno board, and a nRF24L01 (http://www.nordicsemi.com/eng/Products/2.4GHz-RF/nRF24L01) communication module. 

The Arduino Uno microcontroller board contains the ATmega328P (http://ww1.microchip.com/downloads/en/DeviceDoc/Atmel-42735-8-bit-AVR-Microcontroller-ATmega328-328P_Datasheet.pdf) 8-bit microcontroller based on AVR RISC architecture operating up to 20 MHz and embedding 2 kB internal SRAM and 32 kB flash program memory. The nRF24L01 is a single chip 2.4 GHz transceiver with an embedded baseband protocol engine (Enhanced ShockBurst™), designed for ultra-low power wireless applications. It is designed to operate in the worldwide 2.4 GHz ISM frequency band with rates from 250 kbit/s up to 2 Mbit/s. The USR sensor interfaces directly with the analog pins of the microcontroller board. The sampling frequency was set to 1 Hz. Using the Arduino integrated development environment (IDE), the board can be programmed offline allowing the uploading of the different fusion algorithms without the use of an external hardware programmer. To illustrate the performance of a control system based on the DS-HMM architecture, the board was used to control a simple LED. 

### 3.2. Data Log

In order to obtain the real-time system estimation, the estimated likelihood ratio was communicated to the CoolTerm (http://freeware.the-meiers.org/) serial port terminal application. The strings were received as serial data and saved in an Access database. Those values where used to evaluate the system performance in a post-processing step. 

### 3.3. Study Design

We installed the system at a typical student office room at the campus of Eindhoven University of Technology. The two USR sensors were mounted on the top right and left corners of the computer screen, respectively. The sensors covered a field of view of approximately 45o in the horizontal plane. Ranging was set to measure distances up to 150 cm. The actual installation set-up and sensor modalities used are shown in Figure 5. The desk was occupied by a PhD student who volunteered to participate in the experiment. The participant was asked to maintain her usual working style. In order to obtain the ground truth, we used the build-in webcam on a Toshiba Satellite Laptop to record user activity. In a post-processing step, the acquired video recordings were processed to get the actual leave-in and -out times and to label user activity.

The study was divided in four phases, each lasting for 10 days; so, in total, 40 days were recorded. Algorithms 1, 2, and 3 were implemented in phases 1, 2, and 3, respectively. During those phases, the fixed values were used for the transition probability matrix. In the fourth phase of the experiment, Algorithm 3 was combined with the use of dynamic transition probabilities. 

### 3.4. Experimental Results

To experimentally evaluate the performance of the proposed DS-HMM architecture, we carried out the series of experiments described in Section 3.3. The achieved results are presented and discussed below. 

The most important aspect to investigate is system performance. We focused on lighting control as an example that demonstrates the potential reduction in lighting energy consumption. Besides optimizing energy consumption, user comfort continues to be the most essential success criterion for smart building applications. In order to optimize this tradeoff, we needed to consider appropriate performance evaluation metrics. Towards this end, we chose the false negative rate (FNR), i.e., the total time that the algorithm wrongly assumed non-presence normalized over the total presence time, as a metric that reflects user annoyance (when the appliance turns off during the user’s presence). We used the percentage of lighting power consumption compared with the baseline of the manual control as a metric that reflects the energy saving potential of the proposed architecture. As the manual control, we considered a system where users turn on the lights as soon as they enter the room and turn them off when leaving the building but do not do so during short breaks during the workday. Figure 6 demonstrates the performance of the tested DS-HMM solution during the implementation phases. The accuracy of each algorithm is also depicted. The different points correspond to a different choice of presence/absence threshold (δ∈{−2, 2}). The communication threshold was set to γ=1.5. According to our results, all the algorithms appear to be an adequate data fusion method, not too far off from an ideal system with unlimited communication. Significant energy savings were reported without sacrificing user comfort. The use of dynamic transition probabilities significantly improves performance, especially in terms of user annoyance without significantly increasing the computational cost. The algorithm was implemented on a simple Arduino microcontroller using only 27% of the available program storage space (in comparison with the implementation that uses constant transition probabilities, which requires 24% of the available storage space). Transmitting part of the history of past observations (Algorithm 1) also shows good performance. However, this requires a lot of reprocessing that consumes power and may deteriorate response time in a real-time application, especially with the increasing of the number of nodes. It also increases the payload of messages. During the second phase of the implementation, energy savings appeared to be higher; however, this corresponds to the higher absence of the user and should not be ascribed to better performance. This is further indicated by the corresponding accuracy that appears to be lower compared with the rest of the fusion algorithms. 

In addition to the energy savings in the generation of light, it is also important to investigate the communication requirements. Our approach substantially reduces communication between nodes, compared with centralized solutions with unconstrained communication; the communication reduction ranges from 85% to 96%, by far more than an order of magnitude. Hence, we may claim that the proposed solution is able to achieve satisfactory performance while considering the energy and communication constraints of battery-powered sensor nodes. In addition, the proposed DS-HMM architecture shows robustness to the communication limitations imposed by a busy environment with heavy wireless traffic, such as at a university. Our experimental results indicate that even under a real channel with realistic channel impairments and packet failures, the algorithm still maintains robust performance, which makes it an attractive solution for wireless sensor networks (WSNs).

### 3.5. Scalability

The ability to arbitrarily scale up the size of a WSN is important for many applications. This motivated us to test how the accuracy of the distributed solution relates to increasing the number of nodes. Because no experimental data was available to us for a large number of sensor nodes, we generated synthetic test data that were statistically representative and at least intuitively plausible. From our experiments, we obtained a large dataset of USR sensor data that we used as a training set. Particularly, the temporal correlation of errors has a strong impact. Figure 7 shows the conditional probability of a sensor error at time t+i given an error in the previous time instant *t*. Our experimental data confirmed our expectation that sensor errors are highly correlated in time, i.e., a sensor that makes an error at time *t* has a high probability to also make an error at time t+i. This time correlation is represented in the synthetic data by bursts of errors, while we kept the underlying process (room occupancy) as obtained from a real-life process. We generated a series of binary sensor readings (1 representing presence or 0 representing absence) by maintaining a specific (fixed) average error probability for both states. We inserted time correlation into the sensor errors by fitting an exponential distribution to the correlation curve derived from the experimental data. In total, 100 time-series synthetic datasets were generated for 10 nodes. 

Figure 8 shows the accuracy of the distributed HMM (Algorithm 1) as a function of the number of collaborating nodes. As expected, increasing the number of nodes increases accuracy. For a small number of nodes, the performance sharply increases if more nodes contributed, but for larger sizes of the node population, this effect levels off. Surprising is the limited effect that a second node has, while the performance increases sharply with the addition of a third node. In fact, a fusion operation on two nodes with similar accuracy profiles implies that if only one sensor observes presence, the system may be hesitant to accept this as a shared conclusion. Increasing the number of nodes adds more trust to what the majority believes. The results are presented for different communication threshold values (γ) that define the trade-off between accuracy and data exchange. A lower threshold (γ = 1) achieved higher accuracy, but at the cost of increased communication. When communication was less frequent (communication threshold γ = 3), we observed larger deviations. The non-monotonicity is a statistical fluctuation that is more pronounced for a system that too aggressively constrains the number of transmissions.

## 4. Discussion

The simple prototype we built was used to act as an initial proof-of-concept of the feasibility of the idea. Although the experiment is limited to only two USR sensors, the goal of our work is to provide a common platform that allows the combination of multiple sensing modalities. In fact, we expect that system detection performance can be significantly improved by using heterogeneous sensing modalities, each with different error and reliability profiles capabilities, for example a passive infrared (PIR) motion sensor that offers high reliability in detecting motion but fails to detect actual occupancy. It is attractive to complement this with a USR sensor that detects distance (and thus occupancy) but is vulnerable to environmental lighting fluctuations or/and with an acoustic sensor that offers non-intrusive recognition, but its performance largely depends on the environment where this technique is applied. For the latter, detection is more accurate in quiet office buildings than in a noisy supermarket or a restaurant environment. Moreover, when people keep silent so no acoustic signal can be collected, these audio-processing algorithms are ineffective. 

To further highlight the features of our proposed solution, we provide a comparison table with existing building occupancy detection mechanisms in the literature (Table 1). These established approaches include radio-frequency identification (RFID), acoustic recognition, image camera, and CO_2_ sensors. RFID and image cameras are not user-friendly in terms of privacy and security and involve high costs. The detection performance of standalone acoustic recognition or CO_2_ sensor varies with environments and suffers from large uncertainty due to temporal noise or fluctuations. Hybrid solutions that involve the combination of sensing modalities [6,7] offer increased detection performance while maintaining low cost. Our approach, as described above, allows sensor reliability and accuracy to be embedded specifically in the data exchanged. However, such centralized approaches require constant communication between the involved sensors that drains the battery lifetime and congests the network. In contrast, our proposed solution offers the advantages of high accuracy and low-cost design while addressing the power consumption (battery lifetime) and data communication requirements of a wireless sensor node.

## 5. Conclusions

With the proliferation of Internet of Things (IoT) devices and technologies, many applications of smart building control are becoming realistically feasible. In this context, we introduce a distributed sensing HMM (DS-HMM) algorithm for occupancy-based control in a smart building environment, which appears to be a new relatively unexplored research problem. Our distributed architecture uses sensors that each autonomously run an HMM algorithm to make local estimates on the likelihood ratio of the user state (presence or absence). The individual sensors communicate (only) according to a newly proposed efficient communication strategy that is based on the local confidence in the user state and update their estimates according to a collaborative fusion function. For optimum detection, with constrained communication, it neither suffices to send the latest observation nor to share only the log-likelihood ratio. Nonetheless, we found that the latter performs reasonably close to an optimum global HMM but is much more attractive for a WSN, as it avoids its excessive communication requirements. We modeled a non-homogeneous HMM with transition rates that depends on the time of day, because activity patterns change during the day. In order to confirm the feasibility of the suggested DS-HMM solution, the entire system was implemented in a simple prototype system with USR sensors and tested experimentally in a typical office environment. Our results showed a 20–36% reduction in the appliance (mains-powered) energy consumption, compared with the baseline measurements (manual control), while maintaining user comfort. Although we considered the energy savings estimations as realistic, consumption may vary for other office settings that include other installations. While we validated our algorithm with ultrasound sensing, our concept of exchanging likelihood ratios, with or without trace-back, lends itself well to other sensing modalities to fuse data in multimodal systems and to allow sensors to even exchange soft-decisions and reliability information.

## Figures and Tables

**Figure 1 sensors-19-01006-f001:**
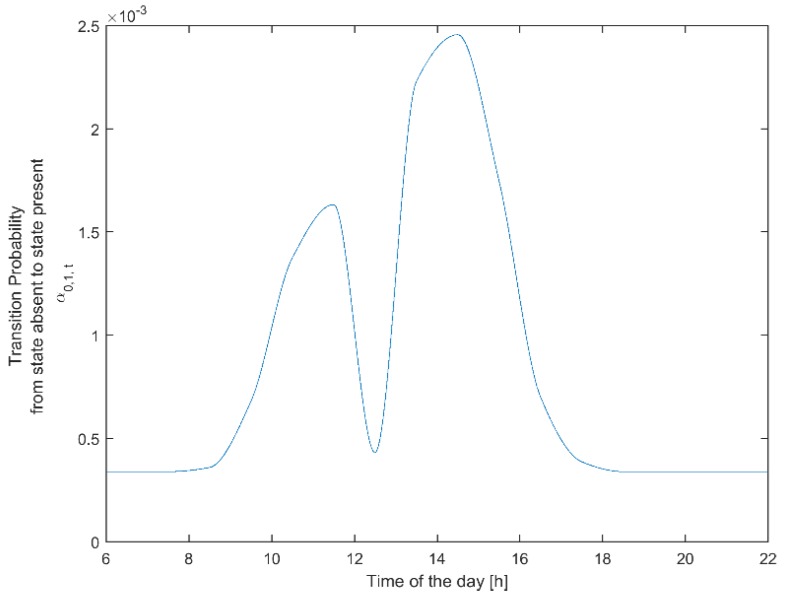
Non-homogeneous transition probability from the absent state to the present state ($a_{01,\;t}=P\left[q_{t+1}=1|q_{t}=0\right]$) as a function of time of day, t.

**Figure 2 sensors-19-01006-f002:**
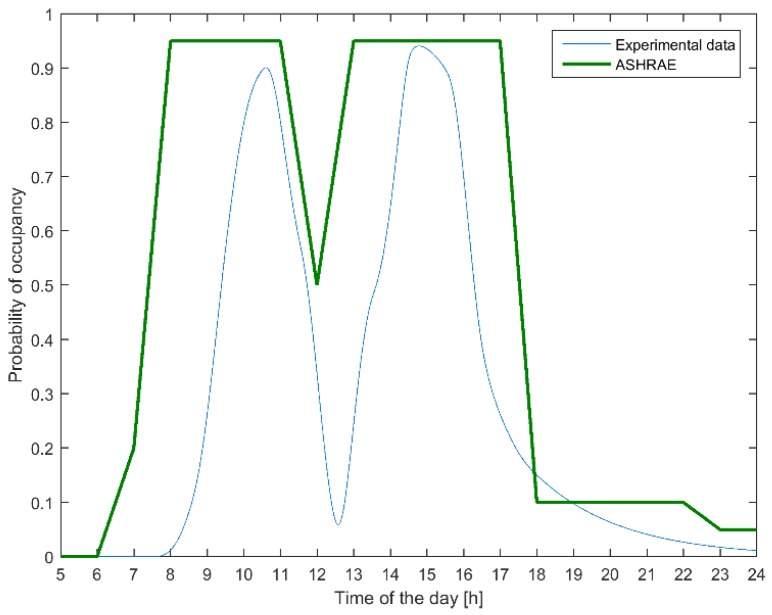
Comparing Occupancy profiles from the ASHRAE 90.1 2004 reference to current study.

**Figure 3 sensors-19-01006-f003:**
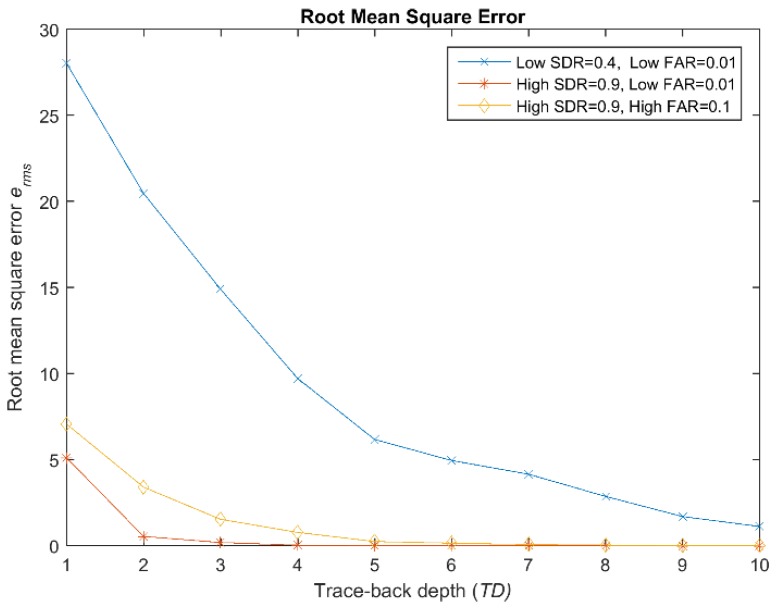
Root-mean-square error $e_{rms}$ as a function of the trace-back depth (TD), i.e., the number of recent observations received.

**Figure 4 sensors-19-01006-f004:**
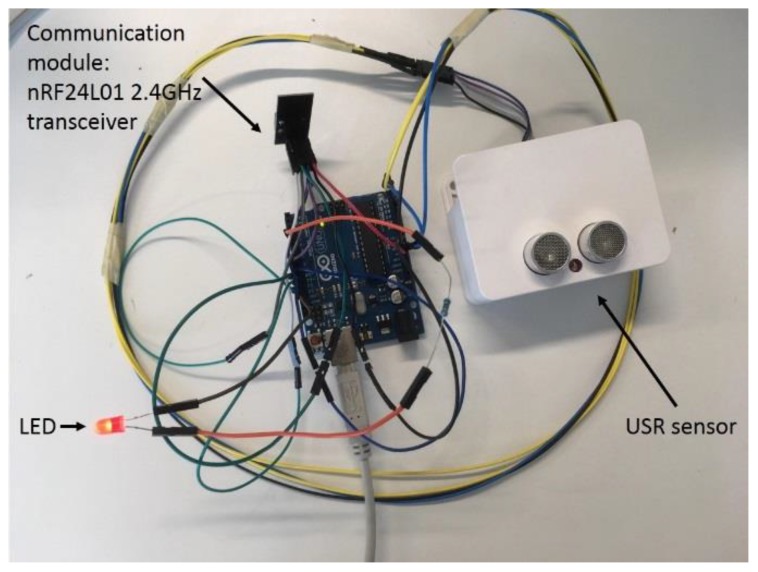
Prototype node realizing the suggested distributed sensing of a hidden Markov model (DS-HMM) architecture.

**Figure 5 sensors-19-01006-f005:**
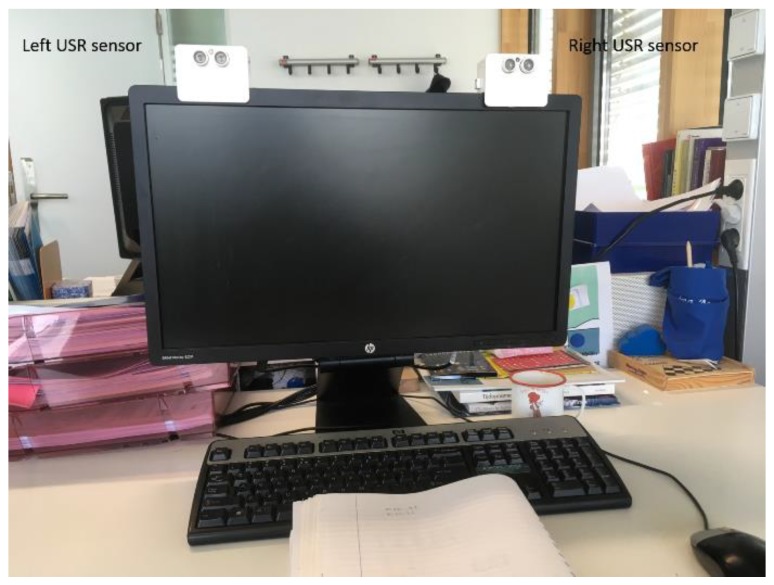
Photo of the set-up installation. Both sensors were mounted at the top edges of the computer screen. A commercial laptop was used to record user activity to obtain the ground truth.

**Figure 6 sensors-19-01006-f006:**
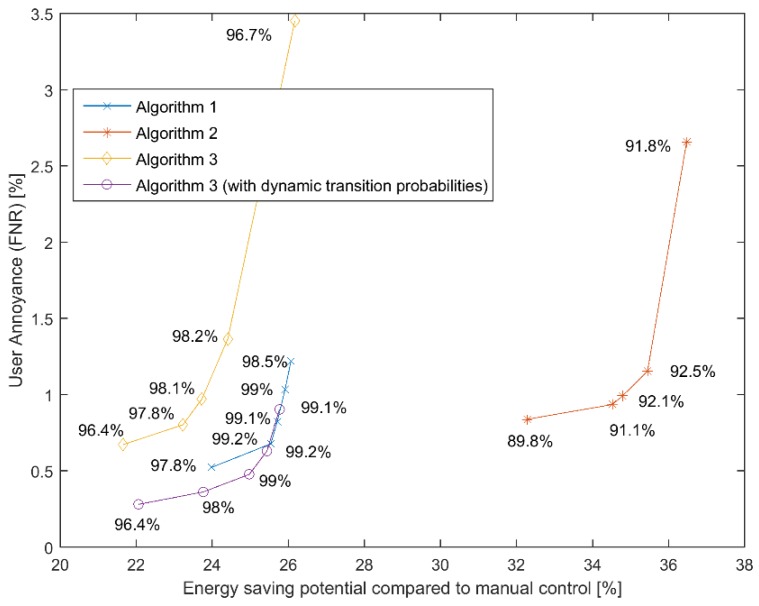
Trade-off between user annoyance, expressed by the percentage of false negatives and energy saving potential during the implementation phases. The accuracy of each algorithm is also depicted.

**Figure 7 sensors-19-01006-f007:**
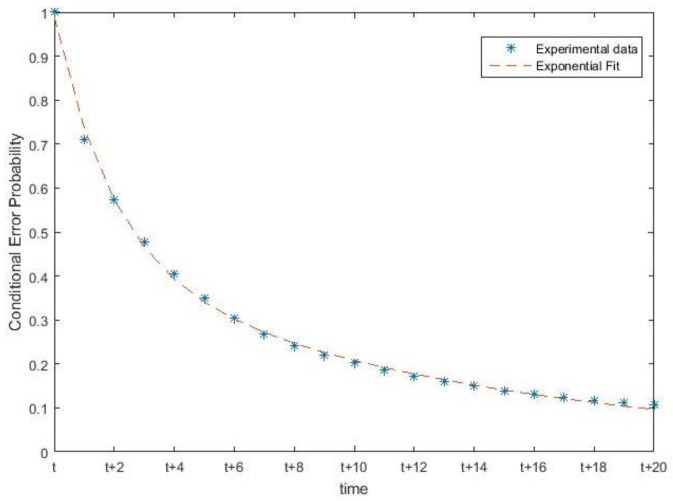
Measured conditional probability of an error at time t+i given an error in the previous time *t*.

**Figure 8 sensors-19-01006-f008:**
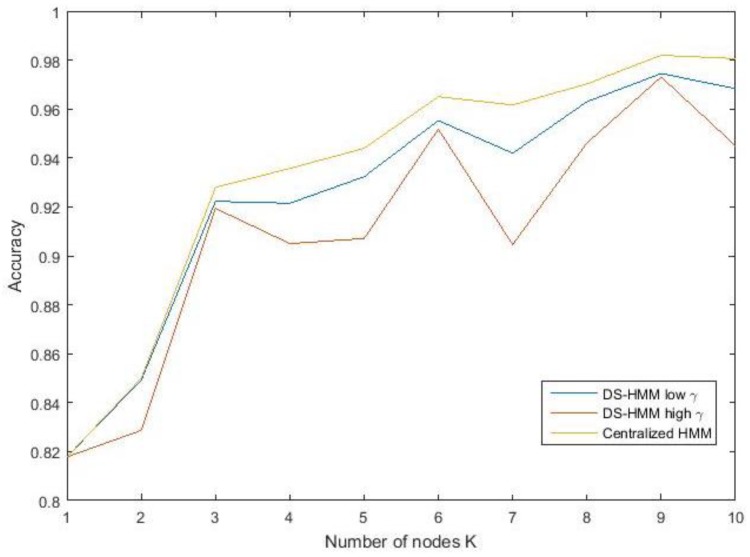
Accuracy as a function of the number of collaborating nodes. The distributed HMM (DS-HMM) solution is compared against a centralized HMM architecture.

**Table 1 sensors-19-01006-t001:** Occupancy detection mechanisms comparison table.

References	Sensing Modality	Processing Algorithm	Cost	Intrusive	Occupancy Detection Performance	Communication Requirements	Centralized (Cloud) or Decentralized EDGE
[3,4]	RFID	SVM Regression models	Low	Yes	High accuracy	Constant connection	Centralized
[2]	Image Camera	Multivariate Gaussian Model	High	Yes	High accuracy	Constant connection	Centralized
[18]	CO_2_	Threshold on sensor reading	Low	No	Accuracy varies by case	Constant connection	Centralized
[19,20]	Acoustic recognition	PCA/LDA Gaussian Mixture Model and HMMs	Low	No	Varying with environment, failure when people keep silent	Constant connection	Centralized
[6,7]	Hybrid	Threshold on sensor reading	Low	No	Improved accuracy with sensor collaboration	Constant connection	Centralized
[8,12]	USR Radar	Centralized HMM			High accuracy	Constant connection	Centralized
This work	USR, but can support any type	HMM	Low	No	High accuracy	Sparse transmissions	Can be decentralized

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
