# Peer review of "Distributed Fusion of Sensor Data in a Constrained Wireless Network"

_sensors, 2019, doi:10.3390/s19051006_

Round 1

Reviewer 1 Report

I think more reference papers are needed, such as

[1] Chen Mao, Qian Huang, Yunfeng Chen, "A compact and versatile wireless sensor prototype for affordable intelligent sensing and monitoring in smart buildings" International Workshop on Computing in Civil Engineering (IWCCE), pp. 155-161, 2017.

          [2] Qian Huang, Chao Lu, Kang Chen, "Smart Building Applications and Information           System Hardware Co-Design," pp. 225-240, Big Data Analytics for Sensor-Network             Collected Intelligence, 2017.

          [3] Chen Mao, Qian Huang, "Occupancy estimation in smart buildings              using hybrid CO2/Light wireless sensor network," ASA Multidisciplineary           Research Symposium, 2016.

2. Please comprehensivly compare and discuss this work with other works in the literature. The purpose is to demonstrate the benefits and disadvantages of the proposed technique with existing works in the literature. 

Author Response

Point 1: I think more reference papers are needed.  

Response 1: We agree that more relevant and updated references improve the paper, so we updated and extended our bibliography.

Point 2: Please compare and discuss this work with other works in the literature. The purpose is to demonstrate the benefits and disadvantages of the proposed technique with existing works in literature.

Response 2: Our solution aims to provide a framework for combining heterogeneous sensing modalities in a distributed architecture. Unlike centralized approaches, that dominate the literature, and that require constant and frequent communication between all nodes, we developed a de-centralized approach based on HMMs and derived fusion formulas that enable to achieve the performance of a centralized solution while addressing the power consumption (battery lifetime) and data communication requirements of a wireless sensor node. So, a comparison mainly highlights these differences in features while maintaining reliable and accurate detection.  We provide a comparison Table covering our and existing building occupancy detection mechanisms in literature.

Reviewer 2 Report

The paper addresses important problems related to the upcoming variety and spread of IoT devices, such as network bottlenecks, communication needs, energy consumption etc. The positive aspects of the papers are: it is well written, the argumentation can be easily followed, the scientific basis in general and the mathematical in particular are sound. I specifically like the development of the HMM, considering dynamics (in regard to daytime deviations of activity) and other aspects - demonstrating the authors' skills in applied research. However, there are some minor weaknesses which should be "repaired": The expectations raised by the abstract / introduction are high - and do not optimally match with the result sections. For example, the relevance of integrating heterogeneous and spatially distributed sensors and actuators (IoT devices) is pointed out - the proof of concept is based on two identical sensors (USR) mounted on a single computer monitor. Although the authors are very transparent with the limitations of the work presented, it should be made more clear that the work is an initial proof of concept /feasibility evaluation of the basic idea, and also enhance the conclusion section in terms of generalizability of results etc. 

A few typos:    Abstract Line 11: expecting ->expected

                              Keywords Line 28: senor -> sensor

                              Line 70: this it - remove "it"

Author Response

Point 1: The expectations raised by the abstract / introduction are high - and do not optimally match with the result sections. For example, the relevance of integrating heterogeneous and spatially distributed sensors and actuators (IoT devices) is pointed out - the proof of concept is based on two identical sensors (USR) mounted on a single computer monitor. Although the authors are very transparent with the limitations of the work presented, it should be made more clear that the work is an initial proof of concept /feasibility evaluation of the basic idea, and also enhance the conclusion section in terms of generalizability of results etc.

Response 1: Indeed, we recognize that the limitations of our prototype need to be presented more clearly, as it aims to act as an initial proof-of-concept and feasibility evaluation of the basic ideas to show functionality and validate the benefits. The main goal was to implement the fusion algorithms on a simple and low cost processor, understand possible bottlenecks in communication and address issues like clock synchronization before proceeding to a further large-scale implementation. Nonetheless, we strongly believe that the underlying mathematical basis for fusing sensor data is sound and that the suggested solution can be generalized to support multiple heterogeneous sensing modalities, and we appreciate that the reviewer recognizes the mathematical soundness. We addressed the concerns, limitations and discussion points in the revised manuscript.

Point 2: A few typos

Response 2: Typos where corrected according to your comments.

Reviewer 3 Report

In IoT systems, local data fusion is required due to the huge volume of sensing data, 

and so distributed architectures become increasingly unavoidable.  

To address this issue, this paper present several methods 

to solve  problems of  Distributed Sensing of a Hidden Markov Model (DS-HMM).

They proposed a set of fusion algorithms to merge data from various HMMs running separately on all sensor nodes.

There are several issues to be addressed for final publication.

1. Most of all, the proof of concept the authors presented in this paper is 

not sufficient to convince readers about their methods.

The implementation setup consists of two sensors for detecting occupancy of the office desk. However, the main objectives of this paper is to solve problems of distributed sensing HMM by minimizing communication among sensors through data fusion. The experiment for the proof-of-concept is overly simplified to reflect the goal of this paper. 

2. Also, the presented results in Section 3 are only about accuracy and energy saving aspects. It would be better that authors present further in-depth analysis of the proposed works such as trends of the accuracy trade-off (or energy saving) with varying sensor numbers or with varying the volume of sensing data.

3. The comparison with previous studies should be presented with simulation or experimental results as well.

Author Response

Point 1: Most of all, the proof of concept the authors presented in this paper is not sufficient to convince readers about their methods. The implementation setup consists of two sensors for detecting occupancy of the office desk. However, the main objectives of this paper is to solve problems of distributed sensing HMM by minimizing communication among sensors through data fusion. The experiment for the proof-of-concept is overly simplified to reflect the goal of this paper. 

Response 1: Indeed, we agree that the implementation set-up is only a simplified prototype. We recognize that the limitations of our prototype need to be presented more clearly, as it aims to act as an initial proof-of-concept and feasibility evaluation of the basic ideas to show functionality and validate the benefits. The main goal was to implement the fusion algorithms on a simple and low cost processor, understand possible bottlenecks in communication and address issues like clock synchronization before proceeding to a further large-scale implementation. Nonetheless, we strongly believe that the underlying mathematical basis for fusing sensor data is sound and that the suggested solution can be generalized to support multiple heterogeneous sensing modalities. We addressed the concerns, limitations and discussion points in the revised manuscript.

Point 2: Also, the presented results in Section 3 are only about accuracy and energy saving aspects. It would be better that authors present further in-depth analysis of the proposed works such as trends of the accuracy trade-off (or energy saving) with varying sensor numbers or with varying the volume of sensing data.

Response 2: We recognize scalability as an important consideration and we have added a new section on this topic, including numerical results, to the extent possible within the time span of 10 days given for revision of the paper. We have addressed this by running simulations on data generated to reflect correlation in measurement data. Our main conclusion is that DS HMM is scalable but that a (short) trace-back is highly recommended to achieve high performance.

Point 3: Also, the presented results in Section 3 are only about accuracy and energy saving aspects. It would be better that authors present further in-depth analysis of the proposed works such as trends of the accuracy trade-off (or energy saving) with varying sensor numbers or with varying the volume of sensing data.

Response 3: Our solution aims to provide a framework for combining heterogeneous sensing modalities in a distributed architecture. Unlike centralized approaches that require constant and frequent communication between all nodes, we developed a de-centralized approach based on HMMs and derived fusion formulas that enable to achieve the performance of a centralized solution while addressing the power consumption (battery lifetime) and data communication requirements of a wireless sensor node. To our knowledge, this is the first time in literature that such a distributed solution based on HMMs is suggested together with the fusion algorithms to implement it. A comparison of the centralized HMM solution with state-of-art classifiers (Naïve Bayes, SVM and Liner Regression) can be found in our previous work in (Papatsimpa & Linnartz, 2017), to which we now refer in the paper. Moreover, according to your suggestions, we recognize the necessity to discuss further and compare this work with existing building occupancy detection mechanisms. To highlight this, we added a comparison Table to demonstrate the benefits and disadvantages of the proposed technique with existing works in the literature. So, a comparison mainly highlights the differences in features while maintaining reliable and accurate detection.

Papatsimpa, C., & Linnartz, J.-P. M. G. (2017). Improved Presence Detection for Occupancy Control in Multisensory Environments. In 2017 IEEE International Conference on Computer and Information Technology (CIT) (pp. 75–80). IEEE. http://doi.org/10.1109/CIT.2017.31

Round 2

Reviewer 1 Report

The quality of this paper has been improved a lot.

By the way, I just read a newly published paper in the Journal of Information Technology in Construction, which is closely related with occupancy basedcontrol in smart building. So put include this paper in the reference list and add it in the table on page 18 for comparison and discussion.

Qian Huang, Kane Rodrigeuz, Nicholas Whestone, Steven Habel, "Rapid Internet of Things (IoT) prototype for accurate people counting towards energy efficient buildings", Journal of Information Technology in Construction, vol. 24, pp. 1-13, 2019.